# FusionSAM: Visual Multimodal Learning with Segment Anything Model

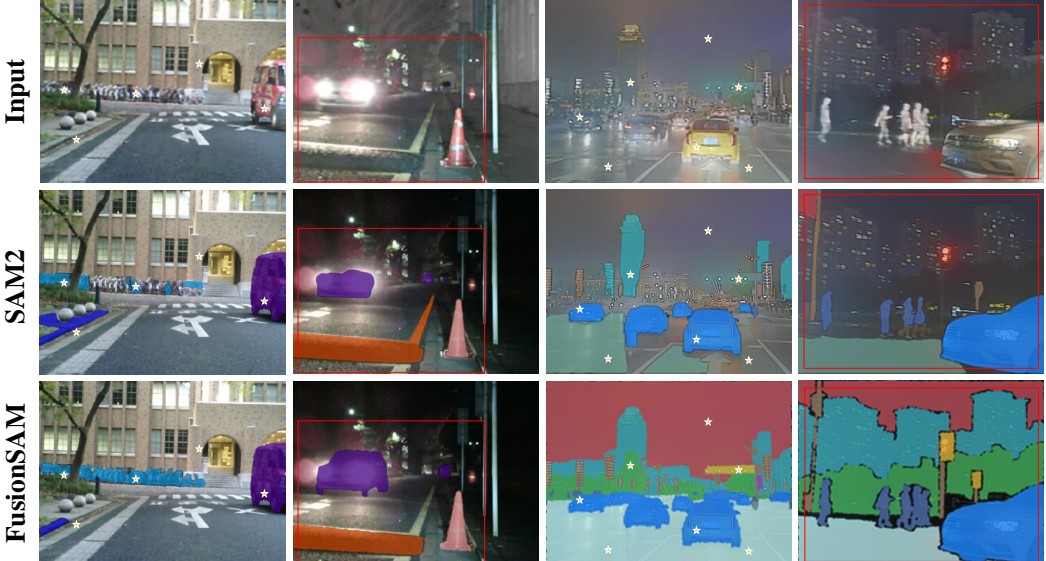

Figure 1: **Comparative results of SAM2 and FusionSAM under the MFNet and FMB datasets:** FusionSAM demonstrates superior boundary accuracy and structural completeness, while SAM2 struggles with misclassifications and unclear boundaries. To ensure fairness, the input is the fusion feature map of our method, annotations for points and boxes prompts are shown in the figure.

## ABSTRACT

Multimodal image fusion and semantic segmentation are critical for autonomous driving. Despite advancements, current models often struggle with segmenting densely packed elements due to a lack of comprehensive fusion features for guidance during training. While the Segment Anything Model (SAM) allows precise control during fine-tuning through its flexible prompting encoder, its potential remains largely unexplored in the context of multimodal segmentation for natural images. In this paper, we introduce SAM into multimodal image segmentation for the first time, proposing a novel framework that combines Latent Space Token Generation (LSTG) and Fusion Mask Prompting (FMP) modules. This approach transforms the training methodology for multimodal segmentation from a traditional black-box approach to a controllable, prompt-based mechanism. Specifically, we obtain latent space features for both modalities through vector quantization and embed them into a cross-attention-based inter-domain fusion module to establish long-range dependencies between modalities. We then use these comprehensive fusion features as prompts to guide precise pixel-level segmentation. Extensive experiments on multiple public datasets demonstrate that our method significantly outperforms SAM and SAM2 in multimodal autonomous driving scenarios, achieving at least a 3.9% improvement in segmentation mIoU over state-of-the-art methods.

## 1 INTRODUCTION

Accurate and comprehensive scene understanding is crucial for autonomous driving (Zhang & Demiris, 2023). Due to the limitations of sensor imaging devices, no single modality sensor can independently provide a complete description of the scene (Zhou et al., 2024; Xue & Marculescu, 2023; Cao et al., 2023b; Xu et al., 2023). For instance, infrared sensors capture thermal radiation information, highlighting objects of interest such as pedestrians and vehicles (Bellagente et al., 2024). Conversely, visible light sensors capture reflected light, generating scenes rich in texture details (Liu et al., 2024). By combining these modalities, complementary details that might be missed by individual sensors can be captured, enhancing the model's ability to perform semantic segmentation of the complete scene (Cao et al., 2023a). Therefore, the fusion of infrared and visible light images has become a mainstream solution for improving scene understanding and semantic segmentation. However, current semantic segmentation models struggle to comprehend densely packed elements in multimodal driving scenes, failing to fully represent the captured information for better subsequent segmentation results.

In recent decades, advancements in semantic segmentation within deep learning have significantly propelled the understanding of multimodal scenes. Capturing efficient multimodal fusion representations is key to enhancing segmentation performance. A common approach involves feature-level fusion of infrared and visible light images using Convolutional Neural Networks (CNNs) to extract rich semantic representations, but the local constraints of CNNs make it challenging to effectively merge information from different modalities. As an alternative, Transformer architectures, with their attention mechanisms and ability to model long-range dependencies, facilitate better global fusion and utilization of complementary information (Li et al., 2023; Zhang et al., 2024). However, pure transformer architectures lack the flexibility required for scene understanding, especially in autonomous driving scenarios where elements are densely packed (Cao et al., 2023c), and edge textures of segmented categories are blurred due to varying lighting conditions and nighttime environments. Without intermediate fine-tuning guidance to focus on critical regions, segmentation distortions can occur, hindering better scene parsing. The Segment Anything Model (SAM) has emerged as a transformative method for single-modal natural scene segmentation due to its flexible prompting architecture (Ravi et al., 2024; Kirillov et al., 2023). Remarkably, the prompt architecture of SAM enhances the model's ability to focus on detailed features. Through the guiding mechanism of prompts, SAM can more effectively direct the segmentation process compared to transformers that lack fine-tuned control. This is crucial for the dense element segmentation required in autonomous driving scenarios. However, SAM has not yet been extensively studied in the realm of multimodal fusion.

To address these challenges, we innovatively propose FusionSAM, a Latent Space driven **S**egment **A**nything **M**odel for Multi-Modal Fusion and Segmentation, which endows SAM with efficient multimodal image fusion and segmentation capabilities. Specifically, we first capture latent space feature embeddings of the two modalities through vector quantization to obtain efficient downsampled representations. Then, we establish long-range dependencies between the modalities using a cross-attention-based inter-domain fusion module, capturing comprehensive information as fusion features to guide precise pixel-level segmentation. To the best of our knowledge, this is the first study to apply the SAM to multimodal visual segmentation tasks in natural images, and it outperforms current state-of-the-art methods as shown in Figure 1. Our main contributions are as follows:

- We extend SAM to multimodal image segmentation in natural images for the first time. Through SAM's flexible prompt encoder we achieve efficient fusion and segmentation of multimodal images, meeting the complex requirements of autonomous driving scenarios with dense elements and varying lighting conditions.

- We propose a novel FusionSAM framework that includes the Latent Space Token Generation (LSTG) and Fusion Mask Prompting (FMP) Module. By capturing latent space representations through vector quantization and performing cross-domain fusion of these features, we generate precise segmentation prompts.

- Extensive experiments on public datasets and benchmarks show that FusionSAM significantly outperforms state-of-the-art methods, including SAM and SAM2, in multimodal autonomous driving scenarios, achieving a notable 3.9% improvement in segmentation IoU, validating its effectiveness and robustness.

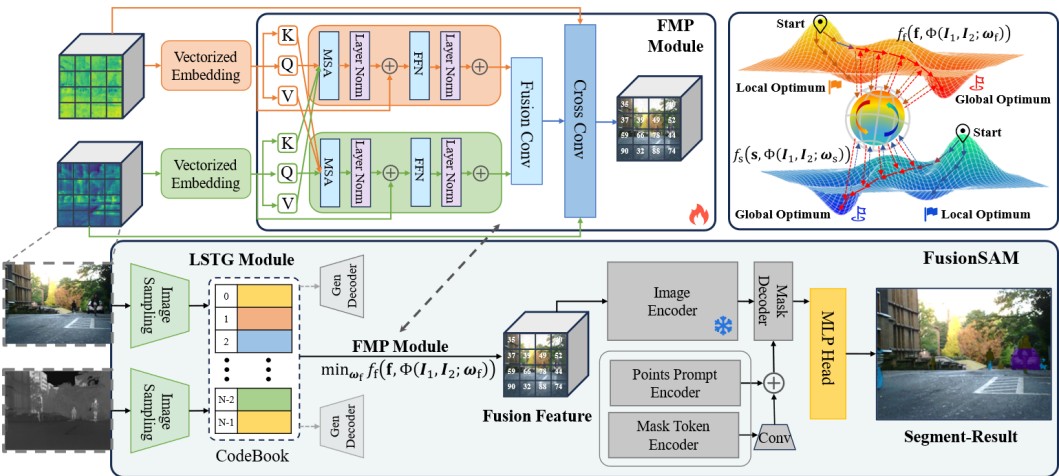

Figure 2: Overview of **FusionSAM** framework for multimodal visual segmentation, which enhances multimodal visual understanding on the original SAM architecture. The main improvements include **Latent Space Token Generation (LSTG)** Module and **Fusion Mask Prompting (FMP)** Module. All parts of the architecture except the image encoder participate in the training phase.

## 2 RELATED WORK

### 2.1 SEGMENT ANYTHING MODEL (SAM)

The SAM enables efficient object segmentation through simple prompt embeddings, like points or bounding boxes, guiding the model to focus on specific regions (Ke et al., 2023; Schön et al., 2024; Ren et al., 2024; Shen et al., 2024a; Wang et al., 2023; Ma et al., 2024; Huang et al., 2024). Derived methods in single-modality segmentation include RobustSAM (Chen et al., 2024) by Chen *et al.*, which improved SAM's performance on low-quality images, and Crowd-SAM (Cai et al., 2024) by Cai *et al.*, which enhanced segmentation in crowded scenes with an Efficient Prompt Sampler and Part-Whole Discriminator Network. SAM has also been adapted for cross-modal tasks in fields like medical imaging and remote sensing. For example, Pandey *et al.* used YOLOv8 and SAM for cross-modal segmentation (Pandey et al., 2023), while Yan *et al.* introduced RingMo-SAM (Yan et al., 2023) for segmenting optical and SAR data. However, these methods only linearly adapt SAM for multimodal tasks, missing the full potential of multimodal features. They also overlook SAM's powerful prompting architecture, which could better activate multimodal fusion features during training to guide segmentation. Our proposed FusionSAM, on the other hand, captures latent space representations through vector quantization, enabling comprehensive cross-domain fusion and using these features as precise segmentation prompts.

### 2.2 MULTI-MODALITY IMAGE FUSION

In autonomous driving, integrating various sensors is essential for accurate scene understanding, as single-modality data is insufficient (Liang et al., 2022; Li et al., 2024; Zhang et al., 2021; Sun et al., 2022; Wang et al., 2020a; 2022; Zhang et al., 2020). Wang *et al.* proposed AsymFusion (Wang et al., 2020b), which enhances multimodal feature interaction using a dual-branch structure with asymmetric fusion blocks. Zhang *et al.* developed MRFS (Zhang et al., 2024), combining CNN-based Interactive Gated Mixed Attention with transformer-based Progressive Cycle Attention to overcome bottlenecks in infrared-visible fusion. Feng *et al.* introduced MAF-Net (Feng et al., 2022), which effectively segments road potholes by fusing RGB and disparity data. Ma *et al.* proposed SwinFusion (Ma et al., 2022), leveraging cross-domain long-range learning and Swin Transformer for global information integration and complementary feature extraction. Most existing methods rely on convolutional networks or transformers, which struggle with global information extraction and flexible segmentation in dense scenes. To overcome these limitations, we apply multimodal fusion within SAM, using its flexible prompting to enhance segmentation in complex autonomous driving scenarios.

# 3 PROPOSED METHOD

## 3.1 PROBLEM FORMULATION

For the task of multimodal image fusion, we first assume visible image $I_1 \in \mathbb{R}^{H \times W \times C_{in}}$ and infrared image $I_2 \in \mathbb{R}^{H \times W \times C_{in}}$, where the two source images from different domains are aligned. Let $H$, $W$, and $C_{in}$ denote the height, width, and channel number of input images, respectively. To achieve pixel-level segmentation, we design an interactive neural network for fusion and segmentation, and optimize the model to find a set of optimal parameters. The optimization model is formulated as follows:

$$\min_{\boldsymbol{\omega}_{\mathrm{f}}, \boldsymbol{\omega}_{\mathrm{s}}} f_{\mathrm{f}} \left( I_f, \Phi \left( I_1, I_2; \boldsymbol{\omega}_{\mathrm{f}} \right) \right) + f_{\mathrm{s}} \left( I_s, \Psi \left( I_1, I_2; \boldsymbol{\omega}_{\mathrm{s}} \right) \right), \tag{1}$$

$I_f \in \mathbb{R}^{H \times W \times C_{\mathrm{in}}}$ and $I_s \in \mathbb{R}^{H \times W \times C_{\mathrm{in}}}$ represent the fusion map and segmentation result, produced by the fusion network $\Phi$ and segmentation network $\Psi$ with learnable parameters $\boldsymbol{\omega}_{\mathrm{f}}$ and $\boldsymbol{\omega}_{\mathrm{s}}$. The functions $f_{\mathrm{f}}(\cdot)$ and $f_{\mathrm{s}}(\cdot)$ correspond to the objective functions for fusion and segmentation, measuring the discrepancies between the predictions and their respective targets.

## 3.2 FUSION SEGMENT ANYTHING MODEL

We propose FusionSAM, which enhances image fusion while preserving the segmentation capability of the SAM architecture. By integrating a fusion module that enables latent space representation embedding and cross-modal consistency fusion into the original SAM architecture, so that its performance will be greatly improved.

### 3.2.1 MODEL OVERVIEW

Figure 2 presents an overview of the proposed FusionSAM. The key contribution of FusionSAM is its Latent Space Token Generation (LSTG) and Fusion Mask Prompting (FMP) modules. Unlike methods that fine-tune or add adapters to SAM and SAM2, FusionSAM's strength lies in its rigorous and well-considered approach to efficient multimodal fusion and segmentation. This efficiency is achieved by fusing compact and comprehensive latent space representations of both modalities, rather than the original large-scale images, enabling more thorough and effective fusion.

**Training.** To train FusionSAM, we first generate efficient fused modality representations, which are then input into the model. Initially, a vector encoder creates latent space representations for both modalities, followed by cross-attention-guided fusion to achieve a comprehensive representation. Unlike the original SAM, we modify the input tokens for segmentation into Full-fledged Output Tokens (FOT), which are enhanced versions of the latent representations designed to capture the full spectrum of fused features for segmentation. These FOTs, along with the prompt token, are processed through the SAM decoding layers to generate the segmentation mask.

The LSTG block processes the raw images from both modalities and transforms them into efficient latent space features. Simultaneously, the FMP module performs multimodal fusion on the obtained latent features. It uses cross-attention mechanisms to learn features from different modality domains, producing refined and comprehensive features. These refined fusion features are then fed into the mask encoder to enhance segmentation quality.

In summary, the robust segmentation capability of the completed FusionSAM framework primarily stems from the training of the LSTG and the FMP modules. Additionally, the decoder and segmentation head from the original SAM architecture are also involved in the learning process. This integration ensures that the model comprehensively understands the fused features from both modalities, thereby enhancing segmentation performance.

**Inference.** In the FusionSAM framework, the ViT-driven image encoder is not involved in training, it is solely used for inference to generate inputs for the mask decoder.

### 3.2.2 LATENT SPACE TOKEN GENERATION

In our multimodal image fusion and segmentation approach, the LSTG module effectively transforms complex input data from visible and infrared modalities into structured latent space representations. This transformation is essential for the efficient integration of diverse information sources.

By drawing inspiration from Vector Quantized Generative Adversarial Networks (VQGAN) (Esser et al., 2021), we enhance our model's capability to capture and fuse complementary features from both modalities, thereby improving the performance of multimodal tasks.

Each image $I_i \in \mathbb{R}^{H \times W \times C}$ is transformed into a spatial set of codebook entries $I_i^q \in \mathbb{R}^{h \times w \times d_c}$, where $i \in \{1, 2\}$, $h = \frac{H}{s}$, $w = \frac{W}{s}$, $d_c$ is the latent dimensionality, and $s$ denotes the scaling factor. This transformation enables the efficient representation of complex multimodal features.

The LSTG module employs an encoder $\mathcal{E}$ to compress the input images into latent vectors, capturing significant features necessary for multimodal integration:

$$z_i = \mathcal{E}(I_i) \in \mathbb{R}^{h \times w \times d_c}. \tag{2}$$

These latent vectors preserve the critical multimodal characteristics needed for subsequent fusion and segmentation, allowing us to efficiently integrate and interpret complementary information from both the visible and infrared domains.

The quantization process translates the encoder outputs $z_i$ into discrete representations using a learned codebook $\mathcal{C}$, aligning and structuring diverse features from both modalities for effective fusion:

$$I_i^q = \text{Quant}(z_i) = \left(\text{argmin}_{c_k \in \mathcal{C}} \|z_{ij} - c_k\|\right) \in \mathbb{R}^{h \times w \times d_c}. \tag{3}$$

By mapping each latent vector $z_{ij}$ to the closest entry in the codebook, the $\text{Quant}(\cdot)$ discretizes the latent representation, this function, aligns similar features from both modalities. This enhances the model's ability to merge complementary information and mitigate modality-specific noise.

The decoder $\mathcal{G}$ reconstructs the original images from these quantized representations, ensuring that the fused representation retains the high fidelity and rich detail necessary for accurate segmentation:

$$\hat{I}_i = \mathcal{G}(I_i^q) = \mathcal{G}(\text{Quant}(\mathcal{E}(I_i))). \tag{4}$$

To optimize the LSTG module for multimodal tasks, we incorporate a reconstruction loss $\mathcal{L}_{\text{rec}}$ to maintain the fidelity of each modality's essential features and a commitment loss $\mathcal{L}_{\text{commit}}$ to ensure effective codebook utilization:

$$\mathcal{L}_{\text{rec}} = \sum_i \|I_i - \hat{I}_i\|^2, \tag{5}$$

$$\mathcal{L}_{\text{commit}} = \sum_i \|\text{sg}[z_i] - I_i^q\|_2^2 + \beta \|\text{sg}[I_i^q] - z_i\|_2^2, \tag{6}$$

where $sg[\cdot]$ denotes the stop-gradient operation. These loss functions help preserve crucial information while promoting the generalization capabilities of the model, which are vital for handling the complexities of multimodal data.

To further enhance the representation quality, a perceptual loss $\mathcal{L}_{\text{perc}}$ and an adversarial loss $\mathcal{L}_{\text{adv}}$ are incorporated. These components focus on maintaining visual coherence and realism across the fused modalities:

$$\mathcal{L}_{\text{perc}} = \sum_i \|\Phi(I_i) - \Phi(\hat{I}_i)\|^2. \tag{7}$$

$$\mathcal{L}_{\text{adv}} = \sum_i \left(\log \mathcal{D}(I_i) + \log(1 - \mathcal{D}(\hat{I}_i))\right), \tag{8}$$

$\mathcal{D}$ is the discriminator network used in the adversarial learning framework, distinguishing between real and generated data. These enhancements ensure that the model captures both low-level detail and high-level semantic information, which is crucial for effective multimodal segmentation. The overall optimization objective combines these elements:

$$\min_{\mathcal{E}, \mathcal{G}, \mathcal{C}} \max_{\mathcal{D}} \sum_i \left[\mathcal{L}_{\text{rec}} + \alpha \mathcal{L}_{\text{perc}} + \beta \mathcal{L}_{\text{adv}} + \gamma \mathcal{L}_{\text{commit}}\right], \tag{9}$$

where $\alpha$, $\beta$, and $\gamma$ are weighting factors that balance the contributions of each loss component, enhancing the model's ability to perform well on multimodal tasks.

The LSTG module's ability to create a robust and structured representation from complex multimodal inputs is key to the successful integration and interpretation of diverse data sources. By

minimizing redundancy while preserving critical information, these tokens facilitate seamless integration into our segmentation framework, significantly enhancing the model's capacity to discern and process complex scenes in multimodal environments. This ensures a comprehensive understanding and efficient handling of the diverse data inherent in visible and infrared images, making the LSTG module a vital component in our multimodal fusion strategy.

### 3.2.3 FUSION MASK PROMPTING MODULE

The FMP module is designed to effectively synthesize latent space representations from visible and infrared modalities, enabling comprehensive scene understanding in autonomous driving scenarios. This module integrates information from different domains presented by each modality into a unified fusion mask. By leveraging the rich and comprehensive features present in the fusion representation as prompts, FMP module provides flexible fine-tuning guidance for the segmentation process, leading to improved segmentation performance. For instance, if multimodal fusion feature map contains complete information, using local area features as point prompts during training can further enhance the model's segmentation accuracy.

Specifically, the FMP module begins with a cross-domain fusion unit that employs cross-attention mechanisms to establish long-range dependencies between different modality domains. This facilitates the exchange of **Queries** (Q), **Keys** (K), and **Values** (V) across domains, ensuring the complete fusion of multimodal features. This process ensures that the fusion mask captures comprehensive interactions between the latent representations $I_1^q$ and $I_2^q$, enhancing the segmentation process by focusing on critical, contextually relevant features that are essential for understanding dense and complex scenes. The inter-domain mechanism is defined as follows:

$$\{Q_1, K_1, V_1\} = \{I_1^q W_{Q1}, I_1^q W_{K1}, I_1^q W_{V1}\},$$
$$\{Q_2, K_2, V_2\} = \{I_2^q W_{Q2}, I_2^q W_{K2}, I_2^q W_{V2}\}, \tag{10}$$

$$z_1' = \mathrm{LN}\left(\mathrm{softmax}\left(\frac{Q_1 K_2^T}{\sqrt{d_k}}\right) V_2\right) + Q_1,$$
$$z_2' = \mathrm{LN}\left(\mathrm{softmax}\left(\frac{Q_2 K_1^T}{\sqrt{d_k}}\right) V_1\right) + Q_2, \tag{11}$$

$\mathrm{LN}(\cdot)$ is the layer normalization, whcih always performed after feed forward network, the outputs $z_1'$ and $z_2'$ represent the globally fused features, which are then processed through a convolutional layer, generating a fused representation $z_c$ that encapsulates the essential information from both modalities. This fused representation serves as the initial fusion mask, guiding the segmentation by highlighting the regions of interest identified through the cross-domain fusion process.

To further enhance the fusion mask, the FMP module integrates a complementary feature fusion unit, which emphasizes the unique characteristics of each modality while ensuring the complete integration of global features. This unit introduces a complementary feature fusion mechanism, where the two modalities are first fused through a cross-attention mechanism to produce $z_0$, which encapsulates the distinctive features of each individual modality. This result is then combined with the initial fusion mask $z_0$, strengthening the segmentation prompt by leveraging the comprehensive information from both approaches:

$$\{Q_0, K_0, V_0\} = \{z_0^q W_{Q1}, z_0^q W_{K1}, z_0^q W_{V1}\},$$
$$\{Q_f, K_f, V_f\} = \left\{z_f^q W_{Q2}, z_f^q W_{K2}, z_f^q W_{V2}\right\}, \tag{12}$$

$$z_f = \mathrm{LN}\left(\mathrm{softmax}\left(\frac{Q_f K_o^T}{\sqrt{d_k}}\right) V_o\right) + z_c. \tag{13}$$

The final representation $z_f$ is then processed through a convolutional layer to produce the fusion mask $I_f$, which serves as a precise prompt for guiding pixel-level segmentation.

By leveraging these cross-domain and complementary feature fusion units, the FMP effectively captures comprehensive fusion features that are critical for accurate segmentation. The integration of global context and long-range dependencies ensures that the model can differentiate between foreground and background elements, even in densely packed autonomous driving scenes. This comprehensive approach allows SAM to achieve robust and high-fidelity segmentation results, effectively addressing the challenges of multimodal image fusion.

The final fused representation $I_f$, derived from the FMP, is fed into the original SAM framework's image encoder. This encoder processes the multimodal fusion results, transforming them into high-dimensional features that encapsulate the rich information from the visible and infrared modalities. The encoded features are then input into the mask decoder, which utilizes a modified transformer architecture to generate mask features through a series of attention operations. Finally, the decoder's output, representing the refined segmentation, is further processed by a multilayer perceptron (MLP) classification head, ensuring that the model accurately identifies and distinguishes between distinct regions within the input data.

## 4 IMPLEMENTATION DETAILS

### 4.1 DATASETS

Two representative datasets, including MFNet (Ha et al., 2017) and FMB (Liu et al., 2023), containing 1569 and 1500 pairs of visible and infrared images with resolutions of 480×640 and 600×800, respectively, to train and evaluate our method. Annotated into 9 and 14 categories relevant to autonomous driving and semantic understanding, these datasets offer varied lighting conditions and rich scenes that enhance the generalization ability of fusion and segmentation models.

### 4.2 TRAINING DETAIL

During 100 epochs of training, multimodal images are subjected to 4× downsampled features by the LSTG module, and the FMP module further captures efficient fusion representations, combined with 10-point mask prompts and 1-box mask prompt to facilitate effective segmentation of SAM. Our initial learning rate is set to 1e-4, using the Adam optimizer with a weight decay of 1e-3, the batch size is set to 4, and vit/h is used as the encoder. All experiments are performed on a NVIDIA A100 Tensor Core GPU. We use mean intersection over union (mIoU) to quantitatively evaluate the performance of semantic segmentation. mIoU is the average result of summing the ratio of the intersection over the sum of the predicted true values for each class.

## 5 EXPERIMENTAL RESULTS

### 5.1 COMPARISONS WITH PREVIOUS METHODS

#### 5.1.1 COMPARISON WITH SAM

SAM (Kirillov et al., 2023) is competitive in the segmentation field because of its powerful segmentation performance and adaptability in different fields. Compared with SAM, SAM2 (Ravi et al., 2024) has significant improvements in applicable fields, segmentation accuracy, and running speed. To demonstrate the effective design and powerful performance of our FusionSAM and maintain a fair comparison, we use SAM and SAM2 to directly infer the fused feature maps generated in FusionSAM, and the results are shown in Table 1. The SAM series cannot handle multimodal image segmentation, whereas our method introduces SAM into the multimodal field, ensuring its excellent segmentation performance and expanding its applicability in more complex scenarios.

| Method | mIoU(%) | | Method | mIoU(%) | |
|---|---|---|---|---|---|
| | MFNet | FMB | | MFNet | FMB |
| SAM | 32.7 | 34.6 | (A) | 35.6 | 41.4 |
| SAM2 | 43.0 | 46.3 | (B) | 47.3 | 57.6 |
| FusionSAM | **63.0** | **61.8** | (C) FusionSAM | **63.0** | **61.8** |

Table 1: Accuracy comparison on two datasets. Table 2: Ablation study results for FusionSAM.

#### 5.1.2 COMPARISON WITH SOTA

We conduct comparative experiments and evaluations with seven state-of-the-art semantic segmentation methods, including EGFNet (Zhou et al., 2022), SegMiF (Liu et al., 2023), EAEFNet (Liang et al., 2023), LASNet (Li et al., 2022), SFAF-MA (He et al., 2023), ECFNet (Shen et al., 2024b),

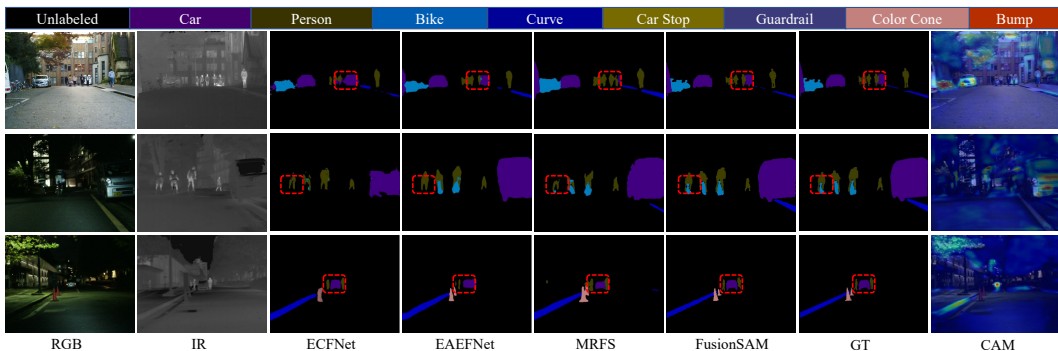

Figure 3: Qualitative demonstrations of different approaches on the MFNet dataset.

| Method | Unlabeled | Car | Person | Bike | Curve | Car Stop | Guardrail | Color Cone | Bump | mIoU(%) |
|---|---|---|---|---|---|---|---|---|---|---|
| EGFNet[23] | 97.7 | 87.6 | 69.8 | 58.8 | 42.8 | 33.8 | 7.3 | 48.3 | 47.1 | 54.8 |
| SegMiF[23] | 98.1 | 87.8 | 71.4 | 63.2 | 47.5 | 31.1 | 0.0 | 48.9 | 50.3 | 56.1 |
| EAEFNet[23] | 97.6 | 87.6 | 72.6 | 63.8 | 48.6 | 35.0 | 14.2 | 52.4 | 58.3 | 58.9 |
| LASNet[23] | 97.4 | 84.2 | 67.1 | 56.9 | 41.1 | 39.6 | **18.9** | 48.8 | 40.1 | 54.9 |
| SFAF-MA[23] | 97.0 | 88.1 | 73.0 | 61.3 | 45.6 | 29.5 | 5.5 | 45.7 | 53.8 | 55.5 |
| ECFNet[24] | 98.0 | 85.7 | 73.5 | 59.7 | 45.7 | 36.7 | 4.0 | 47.4 | 55.1 | 56.2 |
| MRFS[24] | 98.6 | 89.4 | **75.4** | 65.0 | 49.0 | 37.2 | 5.4 | **53.1** | 58.8 | 59.1 |
| **Ours** | **98.8** | **89.8** | 74.0 | **75.8** | **69.6** | **60.2** | 0.0 | 37.8 | **61.4** | **63.0** |

Table 3: Results of quantitative segmentation on the test set of MFNet dataset.

and MRFS (Zhang et al., 2024). We provide quantitative results in Tables 3 and 4. Our FusionSAM achieves the highest mIoU on both datasets. Compared with the second-highest method, Fusion-SAM improves mIoU by 3.9% and 0.6% on MFNet and FMB, respectively. More specifically, for heat-insensitive categories, such as Car Stop, Building, Curve, and Bump, our method achieves significant superiority due to the effective visual quality preservation and enhancement. Overall, these findings confirm that our method achieves SOTA excellence in semantic segmentation.

## 5.2 ABLATION STUDY

To explore the contribution of each part of our method in detail, we designed three scenarios: (A) Omitting the LSTG module compared to our FusionSAM; (B) Removing the FMP module from the fusion process and replacing it with direct concat; (C) Complete FusionSAM. The results of the ablation experiment are shown in Figure 5. We can observe that FusionSAM achieves the best segmentation results on both datasets. As shown in Table 1, in (A) , by removing the LSTG module, we notice that the results drop by 27.4% and 20.4%, respectively, while resulting in poor segmentation results, which shows the effectiveness of the LSTG module in generating latent space tokens through vector quantization. Our fusion method is verified in (B). Without introducing the fusion mask hint, the model has difficulty distinguishing the foreground and background, ignoring the unique and complementary features of each modality, resulting in a decrease in mIoU

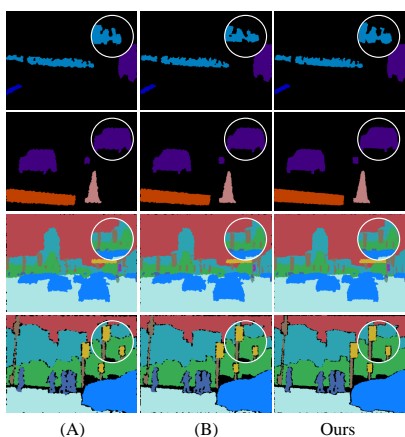

Figure 5: Visualization of ablation studies in FusionSAM.

of 15.7% and 4.2%, respectively. Therefore, our proposed LSTG and FMP module can effectively improve the segmentation performance of multimodal images and produce excellent visual results.

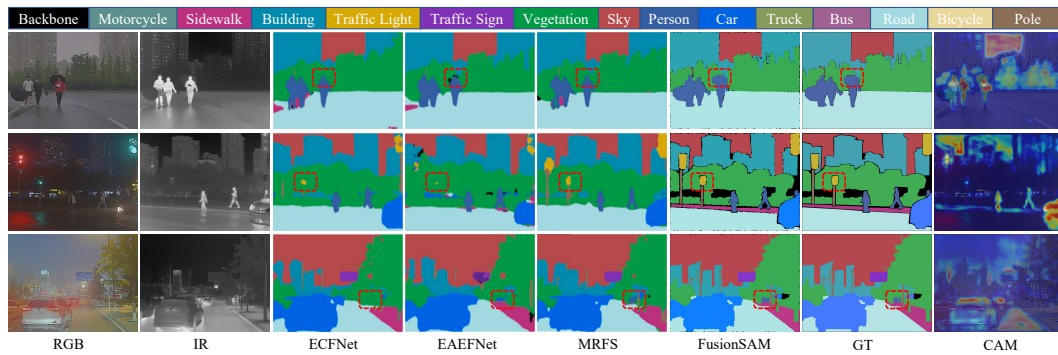

Figure 4: Qualitative demonstrations of different approaches on the FMB dataset.

| Method | Car | Person | Truck | T-Lamp | T-Sign | Building | Vegetation | Pole | mIoU(%) |
|--------|-----|--------|-------|--------|--------|----------|------------|------|---------|
| EGFNet[23] | 77.4 | 63.0 | 17.1 | 25.2 | 66.6 | 77.2 | 83.5 | 41.5 | 47.3 |
| SegMiF[23] | 78.7 | 65.5 | 42.4 | 35.6 | 71.7 | 80.1 | 85.1 | 35.7 | 58.5 |
| EAEFNet[23] | 79.7 | 61.6 | 22.5 | 34.3 | 74.6 | 82.3 | 86.6 | 46.2 | 58.0 |
| LASNet[23] | 73.2 | 58.3 | 33.1 | 32.6 | 68.5 | 80.8 | 83.4 | 41.0 | 55.7 |
| SFAF-MA[23] | 73.0 | 55.7 | 14.3 | 13.6 | 54.2 | 73.0 | 78.9 | 38.1 | 42.7 |
| ECFNet[24] | 80.0 | 63.1 | 12.8 | 40.6 | 71.9 | 81.4 | 84.4 | 44.6 | 52.5 |
| MRFS[24] | 76.2 | **71.3** | 34.4 | **50.1** | **75.8** | 85.4 | 87.0 | **53.6** | 61.2 |
| **Ours** | **80.1** | 52.8 | **45.9** | 43.7 | 46.4 | **85.5** | **88.4** | 50.8 | **61.8** |

Table 4: Results of quantitative segmentation on the test set of FMB dataset.

## 5.3 RESULT VISUALIZATION

Figures 3 and 4 show segmentation visualizations and Class Activation Mapping (CAM) of our method on the MFNet and FMB datasets, and compare with the most competitive methods. These datasets present segmentation challenges due to their rich categories, complex imaging conditions, and diverse scene details. Existing fusion methods struggle to highlight dim infrared targets (e.g., bicycles in Figure 3, second row) and recognize distant pedestrians (Figure 4, third row). Methods relying on two-stream networks often introduce conflicts if feature fusion is incomplete, leading to misclassifications, such as occluded cars (Figure 3, first row) and human shapes (Figure 4, first row). Additionally, edge blurring in dense target predictions is common (Figure 3, third row). By embedding latent space representations and achieving cross-modal consistency, our method reduces redundancy while retaining key information, significantly improving SAM's segmentation performance and enabling accurate object classification across diverse scenes.

## 6 CONCLUSION

A key challenge in multimodal semantic segmentation for autonomous driving is developing a framework that can effectively fuse and utilize multimodal data as prompts during training, guiding the model to achieve high-performance segmentation in dense distribution scenes— an issue that previous multimodal segmentation approaches have not fully addressed. We have innovatively proposed FusionSAM, a latent space driven SAM framework for multimodal semantic segmentation, which endows the SAM architecture with robust capabilities in multimodal fusion, understanding, and segmentation. Our approach performs comprehensive cross-domain fusion of the latent space representations from two modalities, using this fused information as prompts to guide segmentation. This is the first study to leverage SAM in multimodal semantic segmentation of natural scenes, utilizing fusion as a guiding prompt. Extensive experiments demonstrate that FusionSAM significantly outperforms existing state-of-the-art methods in multimodal autonomous driving scenarios, offering a novel approach for future multimodal semantic segmentation tasks.

**Ethics Statement.** This research focuses on multimodal image fusion and segmentation, specifically enhancing performance in autonomous driving scenarios through the FusionSAM framework. The

datasets used, including MFNet and FMB, are publicly available and have been ethically sourced, with proper permissions obtained for their usage. The models developed do not involve human subjects, personal data, or sensitive information, thus avoiding concerns related to privacy, security, or consent. We ensured that no discriminatory or biased data processing practices were employed, as demographic attributes such as race, gender, or other social factors are irrelevant to the model's training and evaluation.

Furthermore, the potential applications of this work are focused on improving the safety and performance of autonomous systems, with no foreseeable risk of harm to individuals or communities. We acknowledge that any technological innovation in autonomous driving carries ethical implications, particularly in terms of safety and responsibility. However, we have ensured that the methods and models developed are aligned with the highest ethical standards in both design and application. Any potential conflicts of interest have been disclosed, and our work adheres to legal and ethical research guidelines.

**Reproducibility Statement.** We have made significant efforts to ensure the reproducibility of all results in this work. Detailed descriptions of the model architecture, including the proposed Latent Space Token Generation (LSTG) and Fusion Mask Prompting (FMP) modules, are presented in Section 3.2.1. The training procedure, including the learning rate, optimizer details, and data used for training, can be found in Section 4.2. Our experimental setup is comprehensively detailed in Section 5, including ablation studies that demonstrate the contributions of each module (LSTG and FMP) to the model's performance. Extensive results on public datasets such as MFNet and FMB are provided in Section 5.2, with quantitative metrics reported for ease of comparison with other methods. Moreover, the exact hyperparameters, dataset details, and the hardware used for training are included in the implementation details (Section 4). Finally, the source code will be made available as anonymous supplementary material, ensuring full reproducibility of the experiments.

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

# 7    APPENDIX

## 7.1    ABLATION VISUALIZATION

As shown in Figure 6, the visualization results from the ablation experiments on the MFNet dataset reveal that the segmentation masks generated without the LSTG and FMP modules exhibit significant jagged edges for objects such as cars and bicycles. This clearly demonstrates the effectiveness of the proposed modules in improving segmentation quality. In contrast, FusionSAM produces smoother and more accurate segmentation masks, highlighting its capability to handle complex scenarios in multimodal image segmentation.

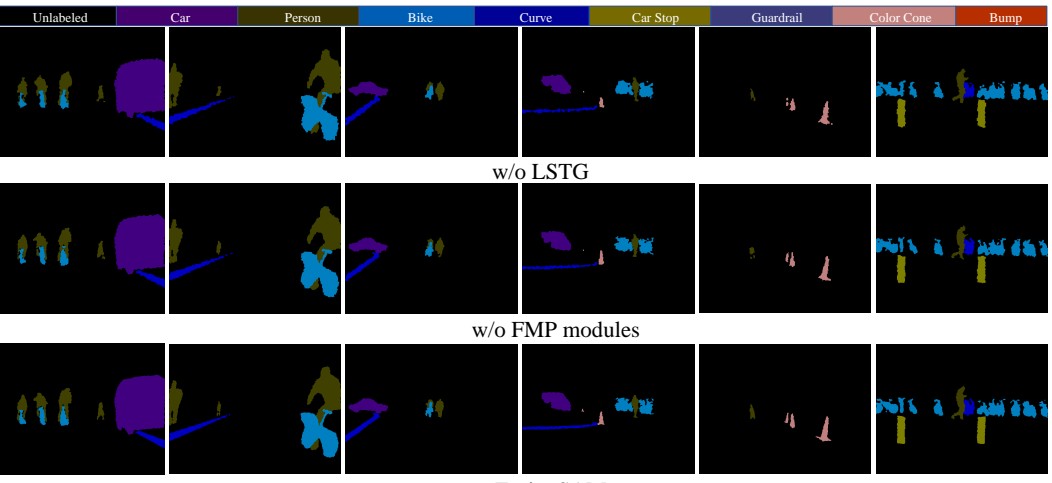

Figure 6: Visualization of FusionSAM ablation research based on MFNet dataset.

Table 5: Ablation results of the MFNet dataset.

| MFNet dataset | | | | | | | | |
|---|---|---|---|---|---|---|---|---|
| **w/o LSTG** | | | **w/o FMP** | | | **Ours** | | |
| **IoU** | **Recall** | **Precision** | **IoU** | **Recall** | **Precision** | **IoU** | **Recall** | **Precision** |
| 97.3 | 99.2 | 98.1 | 97.8 | 99.4 | 98.5 | 98.8 | 99.3 | 99.5 |
| 73.5 | 81.1 | 88.7 | 80 | 86.7 | 91.3 | 89.8 | 95.7 | 93.5 |
| 48.5 | 63.6 | 67.1 | 60.7 | 75.8 | 75.2 | 75.8 | 85.5 | 84.6 |
| 39.1 | 57.4 | 55.2 | 48.8 | 60.4 | 71.7 | 74 | 92.1 | 81.1 |
| 28.7 | 32.1 | 72.9 | 40.9 | 46.6 | 76.9 | 69.6 | 81.4 | 82.7 |
| 13.1 | 15 | 51.5 | 29 | 32.8 | 71.5 | 60.2 | 76.4 | 73.9 |
| 0 | 0 | 0 | 0 | 0 | 0 | 0 | 0 | 0 |
| 4.3 | 4.7 | 34.7 | 27.1 | 29 | 80.3 | 37.8 | 53.5 | 56.3 |
| 16 | 18.6 | 54 | 41.7 | 46.3 | 80.9 | 61.4 | 62.9 | 96.2 |

As shown in Figure 7, the visualization results from the ablation experiments on the FMB dataset show that the segmentation outputs without the LSTG and FMP modules suffer from discontinuities in structures like streetlight poles. In contrast, FusionSAM generates continuous and accurate segmentations, underscoring the effectiveness of the proposed modules. These results further highlight FusionSAM's ability to maintain structural coherence and precision in challenging multimodal segmentation tasks.

The ablation experiment results from the MFNet and FMB datasets, as presented in Tables 5 and 6, demonstrate the superior performance of our proposed method, FusionSAM. For the MFNet dataset, our method significantly outperforms the configurations without the Latent Space Token Generation (LSTG) and Fusion Mask Prompting (FMP) modules. Specifically, the segmentation IoU metrics reveal that while the models without LSTG or FMP struggle with various object classes, our ap-

Table 6: Ablation results of the FMB dataset.

| FMB dataset | | | | | | | | |
| --- | --- | --- | --- | --- | --- | --- | --- | --- |
| w/o LSTG | | | w/o FMP | | | Ours | | |
| IoU | Recall | Precision | IoU | Recall | Precision | IoU | Recall | Precision |
| 52.5 | 54 | 94.9 | 79.3 | 84.5 | 92.9 | 80.1 | 96.1 | 82.7 |
| 25.6 | 32.1 | 56 | 50.4 | 75.1 | 60.5 | 52.8 | 90.2 | 56 |
| 27.4 | 34 | 58.7 | 48.3 | 68.8 | 61.9 | 45.9 | 66.3 | 59.9 |
| 0.5 | 1.2 | 0.8 | 1 | 2 | 2 | 43.7 | 45 | 37.5 |
| 23 | 78.9 | 24.6 | 35.5 | 80.4 | 38.8 | 46.4 | 89.8 | 49 |
| 76 | 87.7 | 85.1 | 83.6 | 88.6 | 93.8 | 85.5 | 87.3 | 97.6 |
| 78.5 | 88.6 | 87.3 | 85.6 | 94.3 | 90.3 | 88.4 | 92.9 | 94.8 |
| 21.5 | 22.4 | 85.6 | 41.9 | 47.7 | 77.4 | 50.8 | 73.5 | 62.2 |

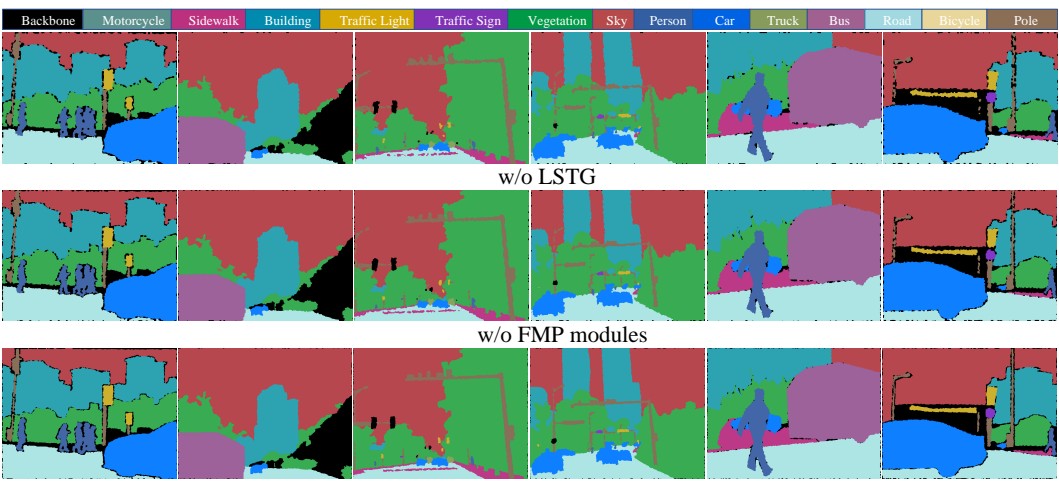

Figure 7: Visualization of FusionSAM ablation research based on FMB dataset.

proach achieves higher precision and recall across the board, with a notable increase in mIoU of at least 3.9%. This indicates that the integration of these modules effectively enhances feature extraction and segmentation accuracy. Similarly, the results on the FMB dataset illustrate the robustness of FusionSAM. The significant improvements in IoU, recall, and precision, particularly in the presence of challenging segmentation tasks, further validate our method's capability to manage dense scenes. For instance, our method consistently produces higher IoU scores compared to the models without the proposed modules, confirming the effectiveness of the LSTG and FMP in enhancing segmentation quality.

Overall, the ablation studies across both datasets highlight FusionSAM's ability to achieve superior segmentation performance, reinforcing its applicability in complex multimodal environments. This robustness positions FusionSAM as a leading method in the realm of multimodal image segmentation for autonomous driving scenarios.

## 7.2 FEATURE VISUALIZATION

In the Feature Visualization section of the appendix, Figures 8 and 9 present the visualization of FusionSAM ablation studies based on the MFNet and FMB datasets. These figures include inputs from both modalities, the ground truth, the fusion results from our FMP Module, feature maps from the segmentation head, and Grad-CAM visualizations of the segmentation outcomes. By examining these visualizations, it is evident that our method effectively integrates the multimodal inputs, leading to improved feature extraction and segmentation accuracy. The FMP Module not only en-

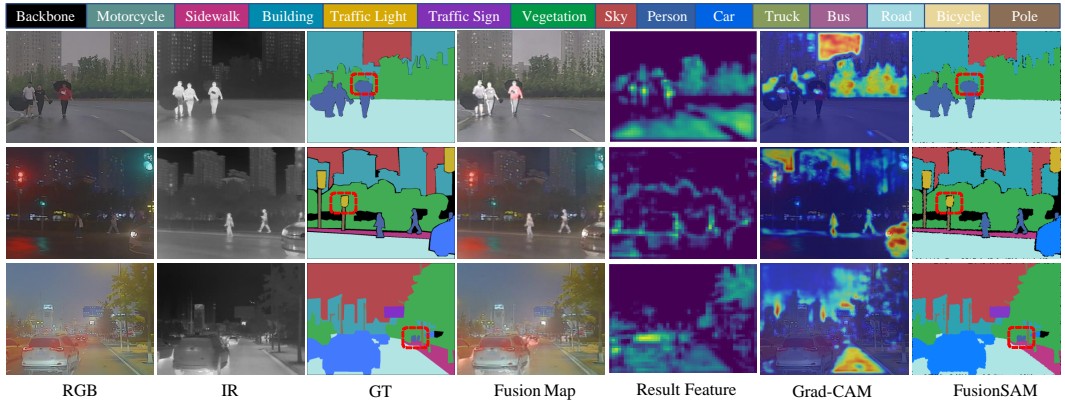

Figure 8: Visualization of FusionSAM ablation research based on FMB dataset.

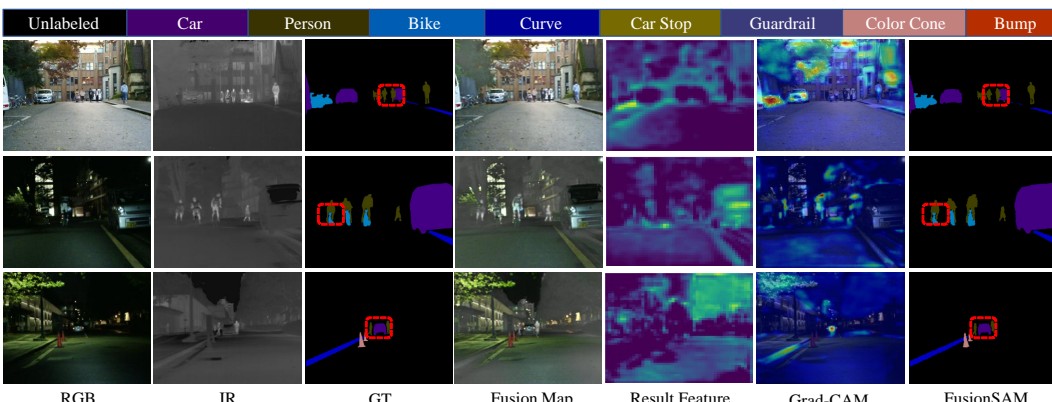

Figure 9: Visualization of FusionSAM ablation research based on MFNet dataset.

hances the fusion of information but also ensures that the segmentation head produces coherent and precise results. Overall, the visual evidence supports the superior performance of our approach in handling complex multimodal segmentation tasks, demonstrating its robustness and applicability in real-world scenarios.

