# OpenReview forum: "FusionSAM: Visual Multimodal Learning with Segment Anything Model"
_ICLR.cc/2025/Conference — ICLR 2025 Conference Withdrawn Submission_

### Official Review · Reviewer_28mf · 2024-10-30

**Soundness:** 3
**Presentation:** 3
**Contribution:** 3
**Rating:** 5
**Confidence:** 4

**Summary:**

The paper introduces FusionSAM, a novel framework that enhances the Segment Anything Model (SAM) for multimodal image fusion and segmentation in autonomous driving scenarios. FusionSAM combines Latent Space Token Generation (LSTG) and Fusion Mask Prompting (FMP) modules to capture comprehensive fusion features from visible and infrared images. These features are then used as prompts to guide precise pixel-level segmentation. The proposed method significantly outperforms existing state-of-the-art approaches, achieving at least a 3.9% higher segmentation mIoU on multiple public datasets. The main contributions of the paper are:

1. Extending SAM to multimodal image segmentation in natural images for the first time.
2. Proposing a novel FusionSAM framework that includes LSTG and FMP modules to enhance multimodal fusion and segmentation capabilities.
3. Extensive experiments have demonstrated that FusionSAM significantly outperforms SAM and SAM2 in multimodal autonomous driving scenarios, validating its effectiveness and robustness.

**Strengths:**

1. The paper introduces FusionSAM, a novel framework that combines Latent Space Token Generation (LSTG) and Fusion Mask Prompting (FMP) modules to enhance the Segment Anything Model (SAM) for multimodal image segmentation. This is the first study to apply SAM to multimodal visual segmentation tasks in natural images, addressing a significant gap in the literature.
2. The authors provide detailed descriptions of the proposed methods, including mathematical formulations and architectural diagrams, which facilitate a clear understanding of the technical aspects.
3. The quantitative results show that FusionSAM outperforms state-of-the-art methods by a notable margin, achieving a 3.9% higher segmentation mIoU than the previous best approach.
4. The paper is well-organized and clearly written. The introduction provides a solid background on the importance of multimodal image fusion and segmentation in autonomous driving.

**Weaknesses:**

1. Lack of comparison with more baseline methods, especially those focusing on multimodal fusion and segmentation.
2. While the paper evaluates the method on two representative datasets (MFNet and FMB), it would be beneficial to include more datasets with varying conditions
3. While the paper provides quantitative results on segmentation performance, it does not analyze the quality of the fused features. Therefore, it should include additional quantitative metrics or visualizations that assess the quality of the fused features.

**Questions:**

* How does FusionSAM perform on datasets with varying lighting conditions and complex backgrounds? Are there any specific strategies to enhance robustness in such scenarios?

* What is the computational cost of the inference process in FusionSAM? How does it compare to the original SAM model?

---

### Official Review · Reviewer_iWKp · 2024-11-02

**Soundness:** 2
**Presentation:** 2
**Contribution:** 2
**Rating:** 3
**Confidence:** 5

**Summary:**

This paper introduces a new multimodal image segmentation framework tailored for autonomous driving, enhancing segmentation in complex, densely packed scenes. Existing models struggle with limited fusion features, while the Segment Anything Model (SAM) offers flexible control, though it's underutilized for multimodal segmentation. The authors propose combining Latent Space Token Generation (LSTG) and Fusion Mask Prompting (FMP) modules to transform training from a black-box method to a prompt-driven system. By using vector quantization and cross-attention, the model generates detailed fusion features to guide accurate pixel-level segmentation. Experiments show that this approach outperforms SAM and SAM2, improving segmentation mIoU by at least 3.9% on public datasets.

**Strengths:**

1.This manuscript is standardized and the writing is fluent, and the content is easy to understand.

2.Experiments on public datasets show that FusionSAM outperforms state-of-the-art methods, including SAM and SAM2, in multimodal autonomous driving, with a 3.9% improvement in segmentation IoU, demonstrating its effectiveness and robustness.

**Weaknesses:**

1. The motivation is unclear and could be better articulated.

2. The paper proposes RGB-infrared semantic segmentation; however, the title refers to "multimodal segmentation," which suggests a broader scope, such as RGB-Depth or RGB-Event segmentation. This mismatch makes the title somewhat confusing.

3. The experimental tables lack details on model complexity, such as FLOPs and parameters. It would also be helpful to include model size and FPS, because the authors use the foundation model SAM.

4. The top-right section of Figure 2 closely resembles Figure 2 from the paper "Multi-interactive Feature Learning and a Full-time Multi-modality Benchmark for Image Fusion and Segmentation."

**Questions:**

Please refer to the concerns and issues raised in the "Weaknesses".

---

### Official Review · Reviewer_JJbN · 2024-11-04

**Soundness:** 2
**Presentation:** 1
**Contribution:** 2
**Rating:** 3
**Confidence:** 5

**Summary:**

The paper introduces SAM into multimodal image segmentation and proposes a framework consisting of the Latent Space Token
Generation (LSTG) module and the Fusion Mask Prompting (FMP) module. The LSTG module encodes the input image into modal-specific latent representation and the FMP module fuses the cross-modal presentation. Experiments on two RGB-T datasets MFNet and FWB were conducted to validate the effectiveness of the proposed method.

**Strengths:**

N/A

**Weaknesses:**

1.**I strongly suspect that this paper was written by an LLM.**

2. The writing is very poor, especially in the method section.
    (1) Why include the top-right figure in Figure 2? There is no explanation in the paper.

    (2) What are the `Fusion Conv` and `Cross Conv` in Figure 2? There is no explanation in the paper.

    (3) In Lines 300-301, `are then processed through a convolutional layer`, how is this achieved?

    (4) What is $z_f^q$ in Equation 12? Which part of Figure 2 corresponds to Equations 12 and 13?

    (5) What are the points prompt and Mask Token in Figure 2? What are the `10-point mask prompts and 1-box mask prompt` in Lines 345-346?

    (6) How is Equation 7 computed?

    (7) Line 288, why are the fused features called `fusion mask`? This is very confusing.

3. What are the differences between the method SAM/SAM2 and FusionSAM in Table 1?

4. The first model in Table 2 omits the LSTG module compared to FusionSAM, so how does it extract latent features for FMP?

5. Line 408, `for heat-insensitive categories, such as Car Stop, Building, Curve, and Bump, our method achieves significant superiority`, however, the improvement on building on the FMB dataset is 0.1 compared with $MRFS_{24}$.  In addition, Why are the improvements on Curve, Car Stop, and Color Cone so large (about 20 points)?

6. More ablation experiments are needed to show the effectiveness of each component in the LSTG and FMP modules, instead of only Table 2.

**Questions:**

See the weaknesses.

---

### Official Review · Reviewer_W1WD · 2024-11-04

**Soundness:** 2
**Presentation:** 2
**Contribution:** 3
**Rating:** 5
**Confidence:** 5

**Summary:**

This paper proposes a multimodal image segmentation framework based on SAM, with Latent Space Token Generation (LSTG) and Fusion Mask Prompting (FMP) introduced. LSTG captures latent space representations through vector quantization and FMP fuses these features through cross-attention. The proposed method is evaluated on MFNet and FMB benchmarks and achieves good results.

**Strengths:**

1. It is the first time to extend SAM to multimodal image segmentation.
2. The authors report good performance on multiple datasets and the improvements are promising.
3. Codes are provided in the supplementary materials for reproducibility.

**Weaknesses:**

1. The design and the involved losses are unintuitive and the authors barely describe the whole pipeline and the proposed components without explaining the motivation and why they have the observed effect. Like this it seems like "we tried it nevertheless and it worked". Here I would have liked to see more theoretical insight.
2. In Table 3, the proposed method obtained 0 mIoU on "Guardrail", which should be a concern to be further analyzed. Why the proposed method fails on the "Guardrail" class specifically? Can the authors propose ideas for addressing this issue?
3. The proposed method only surpasses MRFS slightly on FMB, which makes the proposed method less effective. Please discuss the trade-offs between the proposed method and MRFS, and analyze specific cases where the method performs better or worse. How is the computational efficiency or generalizability compared with MRFS?

**Questions:**

Please explain the motivation for each component when designing the framework and the reason for 0 mIoU on "Guardrail" on MFNet as well as the less effective results on FMB.

---

### Note · Authors · 2024-11-13

I have read and agree with the venue's withdrawal policy on behalf of myself and my co-authors.